

# Synchronizing [10]Be in two varved lake sediment records to IntCal13 [14]C

Markus Czymzik[1,2,3], Raimund Muscheler[3], Florian Adolphi[3,4], Florian Mekhaldi[3], Nadine Dräger[1], Florian Ott[1,5], Michał Słowinski[6], Mirosław Błaszkiewicz[6,7], Ala Aldahan[8], Göran Possnert[9] and Achim Brauer[1]

[1] GFZ-German Research Centre for Geosciences, Section 5.2 Climate Dynamics and Landscape Evolution, 14473 Potsdam, Germany
[2] Leibniz-Institute for Baltic Sea Research Warnemünde (IOW), Marine Geology, 18119 Rostock, Germany
[3] Department of Geology, Quaternary Sciences, Lund University, 22362 Lund, Sweden
[4] Physics Institute, Climate and Environmental Physics, University of Bern, 3012 Bern, Switzerland
[5] Max Planck Institute for the Science of Human History, 07743 Jena, Germany
[6] Polish Academy of Sciences, Institute of Geography and Spatial Organization, Warszawa 00-818, Poland
[7] Polish Academy of Sciences, Institute of Geography and Spatial Organization, Torun 87-100, Poland
[8] Department of Geology, United Arab Emirates University, 15551 Al Ain, UAR
[9] Tandem Laboratory, Uppsala University, 75120 Uppsala, Sweden

*Correspondence to*: Markus Czymzik (markus.czymzik@geol.lu.se)

**Abstract.** Time-scale uncertainties between paleoclimate reconstructions often inhibit studying the exact timing,
spatial expression and driving mechanisms of climate variations. Detecting and aligning the globally common cosmogenic radionuclide production signal via a curve fitting method provides a tool for the quasi-continuous synchronization of paleoclimate archives. In this study, we apply this approach to synchronize [10]Be records from varved sediments of Lakes Tiefer See and Czechowskie covering the Maunder-, Homeric- and 5500 a BP grand solar minima with [14]C production rates inferred from the IntCal13 calibration curve. Our analyses indicate best fits
with [14]C production rates when the [10]Be records from Lake Tiefer See were shifted for 8 (-12/+4) (Maunder Minimum), 31 (-16/+12) (Homeric Minimum) and 86 (-22/+18) years (5500 a BP grand solar minimum) towards the past. The best fit between the Lake Czechowskie [10]Be record for the 5500 a BP grand solar minimum and [14]C production was obtained when the [10]Be time-series was shifted 29 (-8/+7) years towards present. No significant fits were detected between the Lake Czechowskie [10]Be records for the Maunder- and Homeric Minima and [14]C
production, likely due intensified in-lake sediment resuspension since about 2800 a BP, transporting 'old' [10]Be to the coring location. Our results provide a proof of concept for facilitating [10]Be in varved lake sediments as novel synchronization tool required for investigating leads and lags of proxy responses to climate variability. However, they also point to some limitations of [10]Be in these archives mainly connected to in-lake sediment resuspension processes.






## 1. Introduction

Paleoclimate archives provide unique insights into the dynamics of the climate system under various forcing conditions (Adolphi et al., 2014; Brauer et al., 2008; Neugebauer et al., 2016). Particularly the timing and spatial expression of climate variations can provide valuable information about the underlying driving mechanisms

(Czymzik et al., 2016b; Lane et al., 2013; Rach et al., 2014). However, time-scale uncertainties between different paleoclimate records often inhibit the investigation of such climate variations. Climate independent synchronization tools offer the possibility for synchronizing individual paleoclimate archives and, thereby, robust studies of leads and lags in the climate system.

In addition to volcanic tephra layers (Lane et al., 2013) and atmospheric trace gases (Pedro et al., 2011), cosmogenic

radionuclides like $^{10}$Be and $^{14}$C provide such a synchronization tool (Adolphi et al., 2017; Adolphi and Muscheler, 2016). The isotopes are produced mainly in the stratosphere through cascades of nuclear reactions triggered by incident high energy galactic cosmic rays (Lal and Peters, 1967). The flux of these galactic cosmic rays into the atmosphere is, in turn, modulated on up to multi-centennial-scales mainly by solar activity changes. On >500-year intervals further cosmogenic radionuclide production changes induced by the varying geomagnetic field become

increasingly important (Lal and Peters, 1967; Snowball and Muscheler, 2007). Detecting and aligning the externally forced cosmogenic radionuclide production signal via a curve fitting method enables the quasi-continuous synchronization of natural environmental archives (Adolphi and Muscheler, 2016; Muscheler et al., 2014).

One challenge with this approach is the unequivocal detection of the cosmogenic radionuclide production signal because of transport and deposition processes. Subsequent to production, $^{14}$C oxidizes to $^{14}CO_2$ and enters the global

carbon cycle. Varying exchange rates between Earth's carbon reservoirs add non-production variability to the atmospheric $^{14}$C record (Muscheler et al., 2004). This uncertainty can theoretically be accounted for by calculating $^{14}$C production rates using a carbon cycle model. However, changes in Earth's carbon reservoirs are difficult to assess (Köhler et al., 2006). $^{10}$Be in mid-latitude regions is nearly exclusively scavenged from the atmosphere by precipitation (Heikkilä et al., 2013). Varying atmospheric circulation and scavenging during the about one month

long tropospheric residence time (about 1 year stratospheric residence time) result in spatially non-uniform $^{10}$Be deposition patterns (Aldahan et al., 2008; Raisbeck et al., 1981).

To date, synchronization studies based on cosmogenic radionuclides are limited to $^{14}$C records from trees and $^{10}$Be time-series from Arctic and Antarctic ice cores. For example, Adolphi and Muscheler (2016) synchronized the Greenland ice core and IntCal13 time-scales for the last 11000 years. Synchronizing $^{10}$Be records from varved lake

sediments opens the opportunity for the synchronization of terrestrial paleoclimate records around the globe. First studies underline the potential of varved lake sediments for recording the $^{10}$Be production signal (Berggren et al., 2013, 2010, Czymzik et al., 2015, 2016a; Martin-Puertas et al., 2012).

In the following, we attempt to synchronize $^{10}$Be records from varved sediments of Lakes Tiefer See (TSK) and Czechowskie (JC) covering the grand solar minima at 250 a BP (Maunder Minimum), 2700 a BP (Homeric

Minimum) and 5500 a BP with $^{14}$C production rates inferred from the IntCal13 calibration curve (Muscheler et al.,



2014; Reimer et al., 2013). Annual $^{10}$Be time-series from both lake sediment archives yield the broad preservation of the $^{10}$Be production signal during solar cycles 22 and 23 (Czymzik et al., 2015). The targeted three grand solar minima comprise among the lowest solar activity levels throughout the last 6000 years (Steinhilber et al., 2012).

## 2. Study sites

TSK (53°35'N, 12°31'E, 62 m a.s.l.) and JC (53°52'N, 18°14'E, 108 m a.s.l.) are situated within the Pomeranian Terminal Moraine in the southern Baltic lowlands (Fig. 1) (Dräger et al., 2017; Ott et al., 2016; Słowiński et al., 2017). The lake basins are part of subglacial channel systems formed at the end of the last glaciation and have no major inflows, today. Both lakes are of similar size (TSK: 0.75 km$^2$/JC: 0.73 km$^2$), but the catchment of JC (19.7 km$^2$) is about 4 times larger than that of TSK (5.5 km$^2$) (Fig. 1). TSK sediments during the investigated grand solar minima are composed of alternating intervals of organic-, calcite- and rhodochrosite varves as well as intercalated non-varved sections (Dräger et al., 2017). JC sediments for these time windows comprise endogenic calcite varves with couplets of calcite and organic/diatom sub-layers, and an additional layer of resuspended littoral material since about 2800 a BP (Ott et al., 2016; Wulf et al., 2013). TSK and JC are located at the interface of maritime westerly and continental airflow. Mean annual precipitation is similar at both sites, 640 mm a$^{-1}$ at TSK and 680 mm a$^{-1}$ at JC (Czymzik et al., 2015).

## 3. Methods

### 3.1. Sediment sub-sampling and proxy records

Sediment samples for $^{10}$Be measurements were extracted at ~20-year resolution from sediment cores TSK11 and JC-M2015 based on varve chronologies (Dräger et al., 2017; Ott et al., 2016). Complementary sediment accumulation rate (SAR), geochemical X-ray fluorescence (µ-XRF) and total organic carbon (TOC) time-series were constructed using existing high-resolution datasets from the same sediment cores by calculating $^{10}$Be sample averages (Dräger et al., 2017; Ott et al., 2016; Wulf et al., 2016). Measured µ-XRF data (cps) were normalized by dividing by the sum of all elements, to reduce the effects of varying sediment properties (Weltje and Tjallingii, 2008).

### 3.2. $^{10}$Be extraction and AMS measurements

After spiking with 0.5 mg $^9$Be carrier, Be was leached from homogenized sediment samples overnight with 8 M HCl at 60°C. The resulting solutions were filtered to separate the undissolved fractions. Further addition of NH$_3$ and H$_2$SO$_4$ caused the precipitation of metal hydroxides and silicates, which were again removed by filtering. The remaining solutions were treated with EDTA to separate other metals and, then, passed through hydrogen form ion exchange columns in which Be was retained. Be was extracted from the columns using 4 M HCl and Be(OH)$_2$



precipitated through the addition of $NH_3$ at pH 10. The samples were washed and dehydrated three times by centrifuging and oxidized to BeO at 600°C in a muffle furnace. After mixing with Nb, AMS measurements of BeO were performed at the Tandem Laboratory of Uppsala University. Final $^{10}Be$ concentrations were calculated from measured $^{10}Be/^9Be$ ratios, normalized to the NIST SRM 4325 reference standard ($^{10}Be/^9Be = 2.68 \times 10^{-11}$) (Berggren

et al., 2010).

### 3.3. Time-scale synchronization

Lag-correlation analyses were applied to determine best fits between the $^{10}Be$ records from TSK and JC for the Maunder-, Homeric- and 5500 a BP grand solar minima and $^{14}C$ production rates inferred from the IntCal13

calibration curve (Muscheler et al., 2014; Reimer et al., 2013). Before the correlation, all time-series were 75 to 500-year band-pass filtered and normalized by dividing by the mean, to reduce noise and increase the comparability (Adolphi et al., 2014). Significance levels for all correlation coefficients were calculated using 10000 iterations of a non-parametric random phase test, taking into account autocorrelation and trend present in the time-series (Ebisuzaki, 1997). Chronological uncertainty ranges were reported as the time-spans in which significances of

correlations are below the given significant level. Before the analyses, all time-series were resampled to a 20-year resolution.

### 4. Results

$^{10}Be$ concentrations ($^{10}Be_{con}$) were measured in 78 sediment samples from TSK and 73 sediment samples from JC

(Figs. 2 and 3). $^{10}Be_{con}$ in TSK sediments range from 1.13 to 7.09 x $10^8$ atoms g$^{-1}$, with a mean of 3.91 x $10^8$ atoms g$^{-1}$ (Fig. 2). $^{10}Be_{con}$ in JC sediments vary between 0.93 to 3.82 x $10^8$ atoms g$^{-1}$, around a mean of 1.89 x $10^8$ atoms g$^{-1}$ (Fig. 3). Mean AMS measurement uncertainties are 0.12 x $10^8$ atoms g$^{-1}$ for TSK and 0.07 x $10^8$ atoms g$^{-1}$ for JC samples (Figs. 2 and 3).

### 5. Discussion


#### 5.1. $^{10}Be$ production signal in TSK and JC sediments

Environment and catchment conditions can add non-production variations to $^{10}Be_{con}$ records from varved lake sediments (Berggren et al., 2010; Czymzik et al., 2015). To detect and reduce these variations, we perform a three-step statistical procedure following Czymzik et al. (2016a), with a slight modification. First, multi-linear regressions

were calculated between the $^{10}Be_{con}$ records and TOC, SAR, Ca, Si, Ti proxy time-series from TSK and JC, reflecting changes in sediment accumulation and composition (Dräger et al., 2017; Ott et al., 2016; Wulf et al., 2016), to estimate the possible environmental influence ($^{10}Be_{bias}$). Only the TOC and Ca time-series with significant



contributions (p<0.1) for TSK and JC were included to the final multi-regressions. Subsequently, the resulting $^{10}Be_{bias}$ time-series from TSK and JC sediments were subtracted from the original $^{10}Be_{con}$ records in an attempt to construct an environment-corrected version of the $^{10}Be$ record ($^{10}Be_{environment}$). However, this statistical approach also removes variability in the $^{10}Be_{con}$ records only coincident with variations in proxy time-series, but without a

mechanistic linkage, potentially resulting in an overcorrection. Such coinciding variability can be introduced by solar activity variations causing $^{10}Be$ production changes and climate variations imprinted in the proxy time-series. Therefore, final $^{10}Be$ composite records ($^{10}Be_{comp}$) were calculated by averaging the $^{10}Be_{con}$ and $^{10}Be_{environment}$ records from each site. To enhance the robustness of the corrections, the procedure was performed on the complete $^{10}Be_{con}$ records from TSK and JC covering all three grand solar minima. Uncertainty ranges of the calculated $^{10}Be_{comp}$

records are expressed as the differences between the $^{10}Be_{con}$ and $^{10}Be_{environment}$ time-series (Fig. 4).

Calculated $^{10}Be_{comp}$ time-series from TSK and JC sediments yield modified trends, but similar multi-decadal variability as the original $^{10}Be_{con}$ records during the Maunder- (TSK: r=0.84, p<0.01; JC: r=0.91; p<0.01), Homeric- (TSK: r=0.81, p<0.01; JC: r=0.74; p<0.01) and 5500 a BP grand solar minimum (TSK: r=0.89, p<0.01; JC: r=0.68; p<0.01) (Fig. 4). These linkages suggest that our correction procedure predominantly reduced trends in the $^{10}Be_{con}$

records introduced by varying sedimentary TOC and Ca contents, but largely preserved multi-decadal variations connected with varying $^{10}Be$ production (Figs. 2, 3 and 4). Comparable linkages between measured and corrected $^{10}Be$ records (based on a similar approach) were found in Lake Meerfelder Maar sediments covering the Lateglacial-Holocene transition as well as in recent TSK and JC sediments (Czymzik et al., 2015, 2016a).

The close connections to TOC and Ca for TSK and JC might point to depositional mechanisms of $^{10}Be$ in lake

sediment records. Significant contributions to the multi-regression as well as significant positive correlations for TSK (r=0.62, p<0.01) and JC (r=0.77, p<0.01) suggest a preferential binding of $^{10}Be$ to organic material (Figs. 2 and 3). This result is supported by significant positive correlations of $^{10}Be$ with TOC in two annually resolved time-series from varved sediments of TSK and JC spanning solar cycles 22 and 23 and in Meerfelder Maar sediments covering the Lateglacial-Holocene transition (Czymzik et al., 2015, 2016a).

Significant contributions of Ca to the multi-regressions as well as significant negative correlations with $^{10}Be$ for TSK (r=-0.68, p<0.01) and JC (r=-0.62, p<0.01) might point to a reduced affinity of $^{10}Be$ for Ca (Figs. 2 and 3). A similar behavior was detected in studies about $^{10}Be$ scavenging from the marine realm (Aldahan and Possnert, 1998; Chase et al., 2002, Simon et al., 2016).

## 5.2. $^{10}Be_{comp}$ and group sunspot numbers

To evaluate the preservation of the cosmogenic radionuclide production signal based on observational data, the $^{10}Be_{comp}$ time-series from TSK and JC were compared with a group sunspot number record reaching back to 340 a BP (AD 1610) (Svalgaard and Schatten, 2016) (Fig. 5). Since sunspot and cosmogenic radionuclide records reflect different components of the heliomagnetic field (closed and open magnetic flux) no perfect correlation is expected



(Muscheler et al., 2016). Nevertheless, a comparison of a $^{14}$C based solar activity reconstruction with group sunspot data points to a largely linear relationship between both types of data (Muscheler et al., 2016).

Variations in the $^{10}$Be$_{comp}$ records from TSK and JC resemble multi-decadal to centennial variability in the group sunspot number time-series with highest values around the Maunder Minimum (Fig. 5). Secondary $^{10}$Be$_{comp}$ maxima

in TSK and JC sediments broadly coincide with the Dalton Minimum and solar activity minimum around 30 a BP (AD 1920) (Fig. 5). In JC sediments, the $^{10}$Be$_{comp}$ excursion from -50 to 0 a BP (AD 2000-1950) without an expression in the group sunspot number record as well as the about 20-year delayed Maunder Minimum response could be explained by transport of 'old' $^{10}$Be from the littoral to the coring site (see Section 5.3) and/or spatially inhomogeneous $^{10}$Be deposition patterns (Fig. 5).

### 5.3. Synchronizing TSK/JC $^{10}$Be with IntCal13 $^{14}$C

Shared variance of $^{10}$Be and $^{14}$C records can be interpreted in terms common changes in cosmogenic radionuclide production (Czymzik et al., 2016a; Muscheler et al., 2014). Moreover, it provides the opportunity to synchronize cosmogenic radionuclide records from different archives (Adolphi and Muscheler, 2016). Lag-correlation analyses

were performed to synchronize the TSK and JC $^{10}$Be$_{comp}$ records covering the Maunder-, Homeric- and 5500 a BP grand solar minima with $^{14}$C production rates from the IntCal13 calibration curve (Muscheler et al., 2014).

Best fits with IntCal13 $^{14}$C production rates were obtained, when the $^{10}$Be$_{comp}$ records from TSK were shifted by 8 - 12/+4 years (Maunder Minimum; r=0.47, p<0.1), 31 -16/+12 years (Homeric Minimum; r=0.68, p<0.01) and 86 - 22/+18 years (5500 a BP grand solar minimum; r=0.37, p<0.05) towards the past (Fig. 6). All three best fits occur

within the given chronological uncertainties of ± 17 years (Maunder Minimum), ± 139 years (Homeric Minimum) and ± 74 years (5500 a BP grand solar minimum) (Fig. 6). The on average <10-year resolution of the varve-based sedimentation rate chronology for TSK sediments around the Maunder Minimum due to non-varved intervals, does not affect our analyses conducted on records at 20-year resolution.

The best fit between the $^{10}$Be$_{comp}$ record from JC sediments during the 5500 a BP grand solar minimum and IntCal13

$^{14}$C production rates (r=0.81, p<0.01) was determined, when the $^{10}$Be record was shifted for 29 -8/+7 years towards present (within the given chronological uncertainty of ±56 years) (Fig. 6). No significant correlations between the $^{10}$Be$_{comp}$ records from JC and $^{14}$C production rates were obtained for the Maunder- and Homeric Minima, within the respective varve counting uncertainties of ±4 and ±29 years (Fig. 6). This lack of significant correlation might be explained by a change in sedimentation at about 2800 a BP (Fig. 6). Since that time JC varves include an additional

sub-layer of littoral calcite and diatoms transported to the profundal by wave driven water turbulences in fall. Presumably, the resuspended material also contains 'old' $^{10}$Be inhibiting the clear detection of the expected $^{10}$Be production signal (Fig. 6). Since the $^{10}$Be signal present in the resuspended sediments is unknown, this uncertainty is difficult to correct for. Comparable influences of sediment resuspension were also found in an annually resolved





$^{10}$Be record from recent JC sediments indicating anomalous $^{10}$Be values in a varve including a 3.7 mm thick layer of redeposited littoral calcite (Czymzik et al., 2015).

### 6. Conclusions

Detecting and aligning the common cosmogenic radionuclide production variations allows the synchronization of $^{10}$Be time-series from TSK sediments covering the Maunder-, Homeric- and 5500 a BP grand solar minima and JC sediments for the 5500 a BP grand solar minimum to IntCal13 $^{14}$C production rates. These synchronizations provide a novel type of time-marker for varved lake sediment archives enabling robust investigations of proxy responses to climate variations. Mismatches between $^{10}$Be in JC sediments and $^{14}$C production rates during the Maunder- and

Homeric Minimum are likely associated with in-lake resuspension of 'old' $^{10}$Be, altering the expected $^{10}$Be production signal.

### Acknowledgements

MC was financed by a German Science Foundation 'Research Fellowship' (DFG grants CZ 227/1-1 and CZ 227/1-2). Further financial support was provided through an Endowment of the Royal Physiographic Society in Lund, a Linnaeus grant to Lund University (LUCCI) and the Swedish Research Council (Dnr: 2013-8421). FA is supported by the Swedish Research Council (VR grant: 4.1-2016-00218). AA thanks the UAEU for support through UPAR funding. This study is a contribution to the Virtual Institute of Integrated Climate and Landscape Evolution Analyses

(ICLEA), grant number VH-VI-415 and the climate initiative REKLIM Topic 8 'Abrupt climate change derived from proxy data' of the Helmholtz Association. We thank Inger Påhlsson for the extraction of $^{10}$Be from sediment samples. $^{10}$Be data from TSK and JC are available at the PANGAEA data library (www.pangaea.de).

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



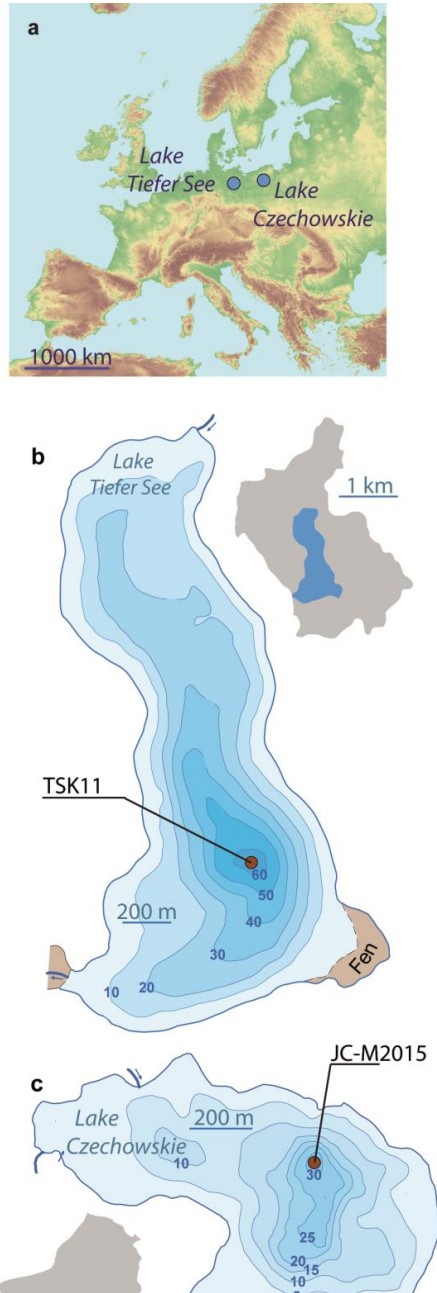

**Figure 1.** Settings of Lakes Tiefer See (TSK) and Czechowskie (JC). (a) Location of TSK and JC in the southern Baltic lowlands. (b) Bathymetric map of TSK with position of sediment core TSK11 and lake-catchment sketch. (c) Bathymetric map of JC with position of sediment core JC-M2015 and lake-catchment sketch.





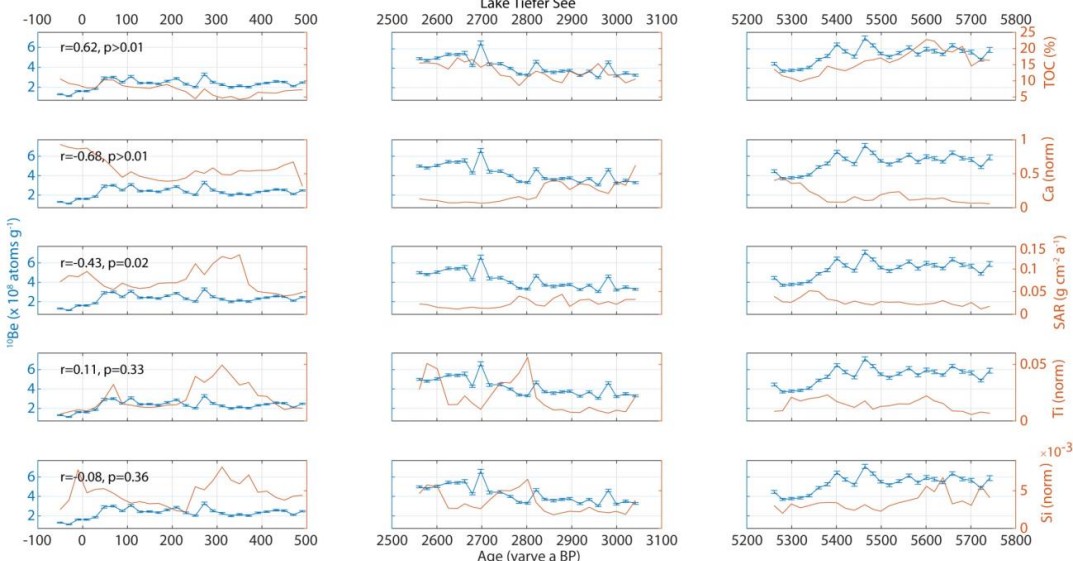

**Figure 2.** $^{10}$Be concentrations ($^{10}$Be$_{con}$) in Lake Tiefer See (TSK) sediments around the Maunder-, Homeric- and 5500 a BP grand solar minima and corresponding proxy time-series from the same archive. $^{10}$Be$_{con}$ compared with sediment accumulation rates (SAR), total organic carbon (TOC), Ti, Ca and Si. Correlation coefficients were calculated for the complete time-series covering all three grand solar minima. Significance levels of correlations were calculated using 10000 iterations of a non-parametric random phase test taking into account trend and autocorrelation present in the time-series (Ebisuzaki, 1997). Error bars indicate AMS measurement uncertainties.





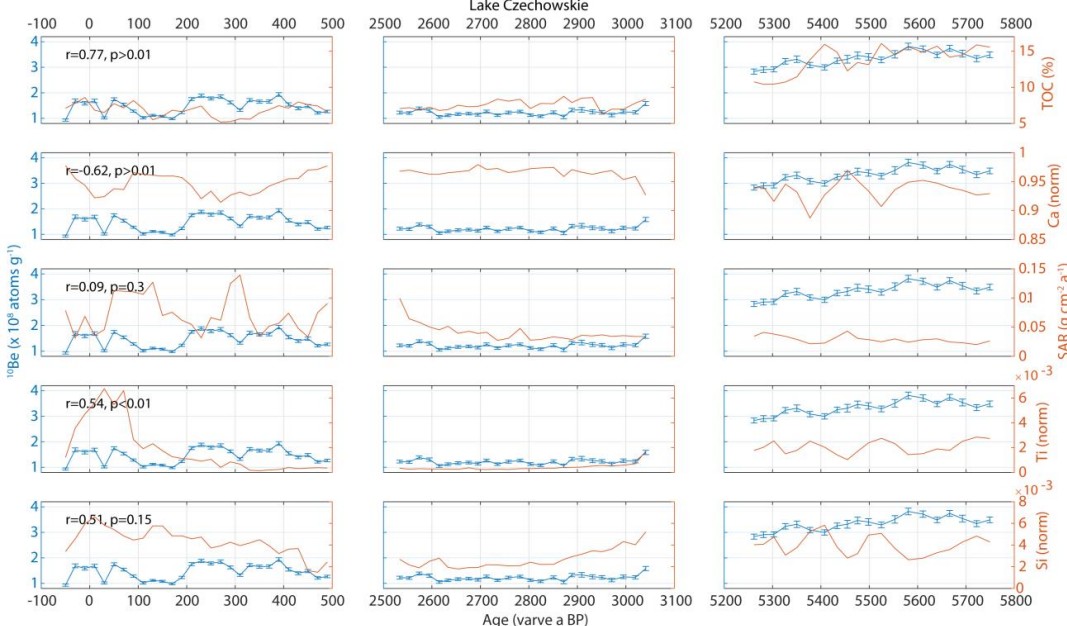

**Figure 3.** $^{10}$Be concentrations ($^{10}$Be$_{con}$) in Lake Czechowskie (JC) sediments around the Maunder-, Homeric- and 5500 a BP grand solar minima and corresponding proxy time-series from the same archive. $^{10}$Be$_{con}$ compared with sediment accumulation rates (SAR), total organic carbon (TOC), Ti, Ca and Si. Correlation coefficients were calculated for the complete time-series covering all three grand solar minima. Significance levels of correlations were calculated using 10000 iterations of a non-parametric random phase test taking into account trend and autocorrelation present in the time-series (Ebisuzaki, 1997). Error bars indicate AMS measurement uncertainties.





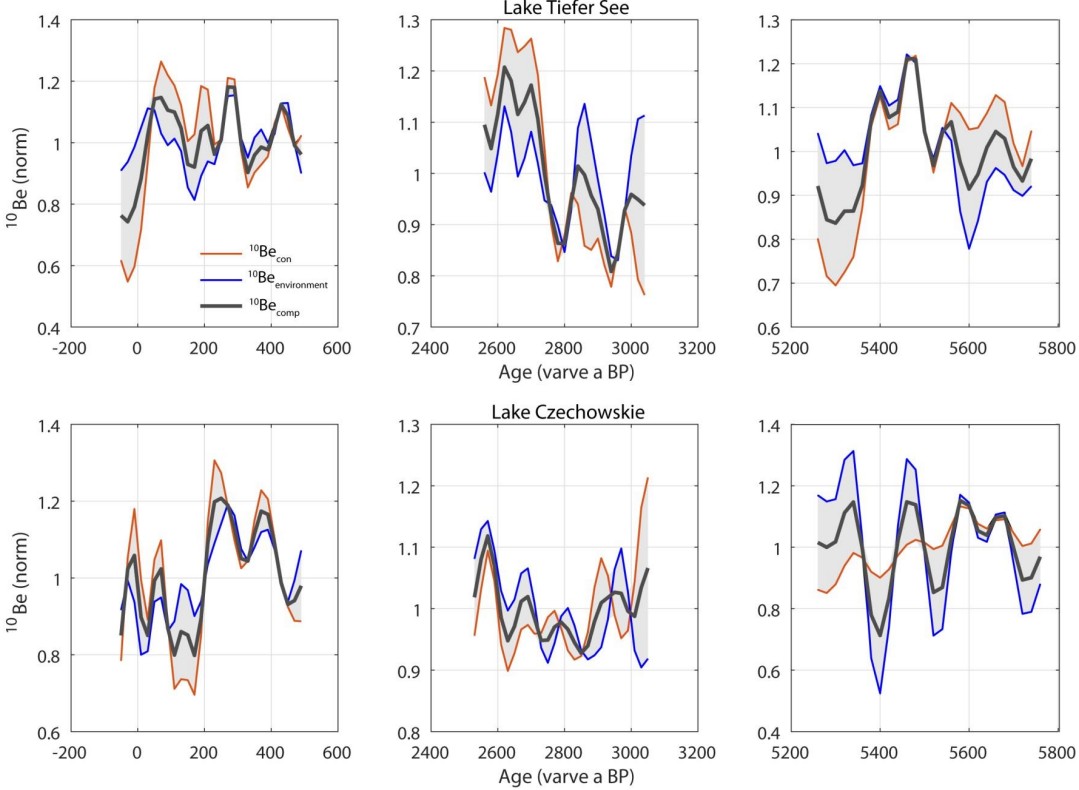

**Figure 4.** Lakes Tiefer See (TSK) and Czechowskie (JC) $^{10}$Be concentration ($^{10}$Be$_{con}$), corrected $^{10}$Be ($^{10}$Be$_{environment}$) and $^{10}$Be composite ($^{10}$Be$_{comp}$) time-series around the Maunder-, Homeric- and 5500 a BP grand solar minima. All time-series are resampled to a 20-year resolution and normalized by dividing by the mean. A 75-year low pass filtered was applied to reduce 5   noise. Uncertainty ranges of $^{10}$Be$_{comp}$ (gray shadings) are expressed as the differences between the $^{10}$Be$_{con}$ and $^{10}$Be$_{environment}$ time-series.





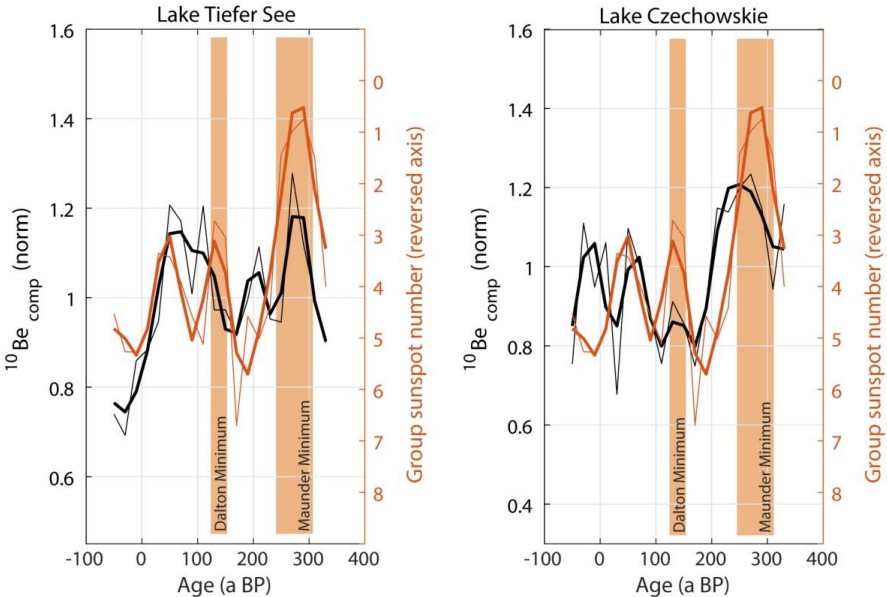

**Figure 5.** $^{10}$Be composites ($^{10}$Be$_{comp}$) from Lakes Tiefer See (TSK) and Czechowskie (JC) compared with group sunspot numbers back to 340 a BP (Svalgaard and Schatten, 2016). Time-windows of the Maunder and Dalton solar minima are highlighted (Eddy, 1976; Frick et al., 1997). Time-series are shown at 20-year resolution (thin lines) and with a 75-year low-pass filter, to reduce noise (thick lines). $^{10}$Be$_{comp}$ records are normalized by dividing by the mean.




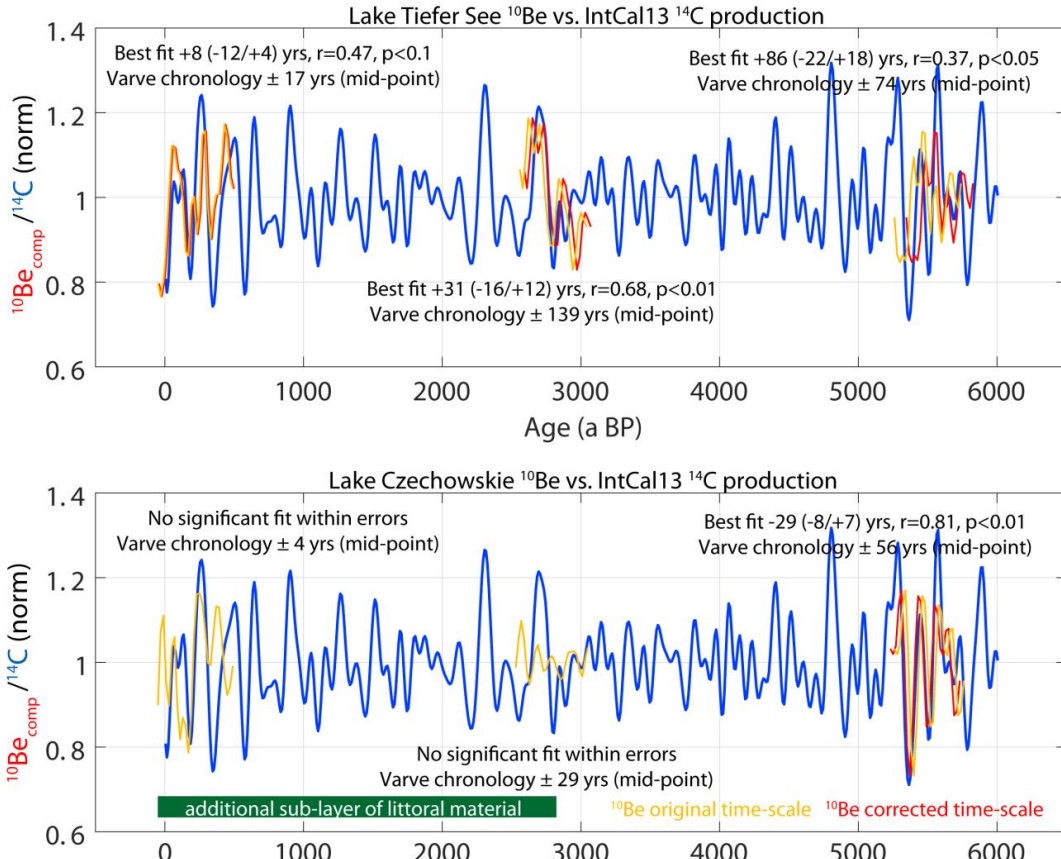

**Figure 6.** Synchronization of $^{10}$Be composites ($^{10}$Be$_{comp}$) from Lakes Tiefer See (TSK) and Czechowskie (JC) for the Maunder-, Homeric- and 5500 a BP grand solar minima with $^{14}$C production rates from the IntCal13 calibration curve (Muscheler et al., 2014). Best fits between the records were calculated using lag-correlation, based on given chronological uncertainties (Dräger et al., 2017; Ott et al., 2016). Significances of the correlations were calculated using 10000 iterations of a non-parametric random phase test taking into account autocorrelation and trend present in the time-series (Ebisuzaki, 1997). Uncertainies are given as the time-spans in which the significances of the correlations are below their respective significant levels. $^{10}$Be$_{comp}$ is shown on its original time-scale and, if applicable, synchronized with IntCal13 $^{14}$C production rates. An interval with additional sub-layers of resuspended littoral material in JC varves is indicated.