# Peer review of "Synchronizing 10Be in two varved lake sediment records to IntCal13 14C during three grand solar minima"

_Climate of the Past, 2017_

## Referee Comment (RC1) · Q. Simon (Referee) · 17 Nov 2017

In this paper, the authors address an essential issue in any paleo-studies, the chronology. They propose a method to help synchronizing varved lakes with other natural archives which is essential to improve our understanding of the mechanisms driving climatic variability. The group of authors already introduced this method for other sites or other time periods in several publications and pursued successfully their investigations in this paper that I recommend for publication following some revisions. The aim of the paper is essentially methodological and I imagine that climatic discussion based on precise inter-correlation of TSK (less evident for JC) with other records, using the 10Be method, will be presented elsewhere (which is fair). The text would greatly benefit from several complementary notes on method, interpretation and discussion (see

comment below). Given the rather short length of the present manuscript, it should be possible to provide these information without weighing too much on the final version of the paper. I listed below a series of comments and questions – voluntarily naïve or not – for which answers (integrated into the paper) should improved robustness an easy readability of the paper.

I would also suggest a slight change of the title to enclose all aspects of the paper: "Synchronizing 10Be in two varved lake sediment records to IntCal13 14C during grand solar minima" (see below).

Comments (page.line):

2.5. The authors could explicitly mention all type of archives which (will) benefit from 10Be for global synchronization. What is the range of time-scale uncertainties associated with these different archives? This is particularly important since it implies different resolutions associated with inherent archive limitations. Despite the most robust archive-to-archive correlation possible (maybe provided by 10Be), these restrictions constitute a limiting factor for studying specific climatic mechanisms in some archives and/or from older ages, particularly about precise lead and lags in the climate system.

2.10. The authors can add paleomagnetism to the series of useful synchronization tools independent from climatic cycles. Use the term "radionuclides" rather than "isotopes".

2.15. Recent works of groups from, e.g., France (Ménabréaz, Valet, Simon. . .) or Japan (Suganuma, Horiuchi. . .) also documented geomagnetic field forcing on the 10Be production variation, is there an impact of these modulation on your records? More largely, what is the impact of solar activity and geomagnetic intensity variations on the magnitude of atmospheric 10Be production rates? Since authors are discussing a synchronization tool that can (will) be used for other time periods, presenting these elements is important because they explain why and how 10Be works, particularly at certain period of time.

2.25. What is the results of this spatial heterogeneity? Does it complicate easy inter-regional correlations? If yes, to what extent? This is important for using 10Be as an accurate global synchronization tool of course.

2.30. Add the recent Raisbeck et al. paper (Clim. Past, 13, 217–229, 2017) which discusses synchronization between Greenland and Antarctic ice cores using 10Be. Does "synchronization of terrestrial paleoclimate records around the globe" need to assume a global homogenization of 10Be production/deposition (see above)? 2.35. Why only studying these three periods? Do you expect higher level of 10Be changes during these intervals? It might be interesting to give some precision here. Moreover, it could be a good idea to mention the three grand solar minima it in the title since your study is focused on these periods.

3.15. What is the extent of sedimentary changes in both cores through the studied intervals? Are they related to any known (studied) climatic cycle? This is important since sedimentary changes can drastically disturb Be records in geological archives. For instance, the last two sentences dealing with current air masses and precipitations behavior are interesting for modern settings but do these parameters also prevailed during the periods scrutinized here?

3.20. To what range of depth intervals correspond a 20-year resolution? What is the sediment amount needed for method? How many years are integrated by the sampling (thickness of the sediment samples)? Also, I do understand that authors want to keep short, which is definitely not a bad idea, but since the chronology is central in the paper (e.g. Title) I find important to present how age models have been obtained (not simply referring to the original publications). What is their resolutions and uncertainties? There is no need to develop too far, but to provide with enough elements for the readers to judge the resolution and potential bias induced by inevitable age errors. This is particularly important since the paper discusses about age offsets with resolutions of only few years back to > 5 ka BP.

3.30/4.5. How do you homogenize sediment samples? What is the sediment weight used? Authors should write that they are interested only by the fraction adsorb or precipitated on sediments (sometimes called "authigenic"), and why are they interested by this fraction? They could precise that metal hydroxides and silicates are precipitated while Be remains in solution. Why precipitate at pH 10 and not 8.5? Are you not precipitating (or risk to precipitate) Boron at this pH level? These last two questions are probably not interesting for the paper, personal interest about the method. It could be useful to add a citation that provide with full description of the method followed here. Add at the end of the last sentence: "and corrected for radioactive decay (Chmeleff et a., 2010; Korschinek et al., 2010)". I totally understand that this correction does not change your results, but better be precise with radioactive elements. Retrieving the exact 10Be concentrations imply a correction for radioactive decay, even if changes occur at the margin given the sediment ages and the T1/2 of 10Be. Note also that authors could add somewhere in the text the half-life of both 14C and 10Be to give the time extent, and therefore theoretical limits, of these tracers (probably more useful for 10Be than for 14C which is already well known by the community).

4.10/15. What is the time uncertainty associated with your data?

4.20. I would remove any mention to Figs. 2 and 3 in the results section as these figures are plotted versus age. Results versus ages are already part of a discussion because they imply a serious transformation through the application of age modeling. Presentation of the raw 10Be concentration data versus depth in new figures is maybe not mandatory since I guess these data will be available as supplementary material or easily available from the web. I know this comment is annoying but discussion will likely evolve while the data will remain, and are therefore important for the community. The authors should highlight directly on the figures the location and duration interval of the grand solar minima discussed (which do not represent the whole box intervals).

4.30. What kind of non-production forcing parameters can explain part of the 10Be concentration variations in varved lake sediments? 2-3 sentences could help readers

to rapidly understand such processes without having to refer to a third party (interested readers will of course go to these citations). I'm wondering why you selected these parameters specifically (i.e. TOC, SAR, Ca, Si, Ti)? Are their fluctuations representing correctly all lithological changes observed in the lakes (e.g. productivity, grain-size, mineralogy)?

5.5. I agree that significant contribution of TOC and Ca on the whole intervals justify their used to the multi-regressions treatment. Yet, it is possible that other elements presented in the paper also impact the 10Be signal within specific depth intervals (e.g. Ti since about 200 a BP in TSK). If they are associated with specific events linked to rapid climatic changes, how can you estimate their residual influence on the 10Been-vironment record calculated? Actually authors correctly discuss that matter later in the paragraph but it results into a blurry questioning about the reliability of the environmental correction procedure, essentially because the method does not rely on any mechanistic linkages between 10Beconc and TOC/Ca, as mentioned by the authors themselves. One could mention here that the method is mainly working because the outcome (10Becomp) is highly comparable with 14C production (Fig. 6) but, although valid, this argument is slightly circular. The main question remains: how to correctly remove, or say diminish, environmental variability imprints on 10Be records in lakes?

5.15. As the authors are interested by multi-decadal variations (see Figures 5 and 6), why not working on 10Beconc series directly as this variability is similar between both 10Becomp and 10Beconc series. This would avoid unnecessary and questionable data treatments while preserving the conclusion.

5.25/25. These two paragraph are rather interesting but could be move above (5.5) to support the use of these two elements for the multi-regression method used to obtain the 10Bebias. Also, it would be interesting to discuss a little bit more (or cite references?) about the exact – or supposed – mechanisms explaining "preferential binding of 10Be to organic material", while the affinity of 10Be to Ca has been indeed demonstrated in several studies already cited in the paper.

6.5. Does result differs when using 10Beconc instead of 10Becomp (see comment above)? In TSK, the unfiltered two 10Be peaks visually correlated with the two sunspot number lows, why not mentioning it? Do you have sedimentological elements to sustain a transport of "old" 10Be? By which processes such a transport can take place (physical remobilization or desorption form sediments previously deposited onto "shelves")? It might be interesting to mention it here, or to refer to explanations provide later in the text.

6.20. Are you using this best fit result to propose a new chronology for TSK?

6.30. See above (point 6.5).

7.5/10. Conclusion is fine and clearly wrap up the main objective of the paper, i.e. 10Be is a robust tool for synchronization (TSK) unless environmental imprint is too strong (JC).

Please also note the supplement to this comment:
https://www.clim-past-discuss.net/cp-2017-117/cp-2017-117-RC1-supplement.pdf

---

## Author Comment (AC1) · 16 Jan 2018

Response to the reviewers' comments We thank Quentin Simon for his constructive and very detailed comments which helped to significantly improve our manuscript. In the following, we will give a detailed response to all concerns that have been raised, first answering the main points of criticism, followed by a point-by-point reply.

(1) Changes in sediment composition, their possible effects on 10Be deposition and how to correct for them

Apparently, we were not clear enough about the possible effects of changes in sediment composition on 10Be deposition in TSK and JC, and how we try to reduce them. Therefore, we modified the manuscript in two ways: (1) To improve the overview on

varying 10Be concentrations and sediment composition in TSK and JC, we now show in addition to Figs. 2 and 3 (10Be concentrations, Ti, SAR, TOC, Si and Ca proxy time-series vs. time) our 10Be records against sediment core depth in the new Figure S1, as suggested by Quentin Simon. (2) To be clearer about the possible modification of 10Be through changes in sediment composition as well as our correction procedure, we have extended and restructured Chapter 5.1, following the reviewer's comments. We now first present our statistical approach (multi-regression analyses) applied to detect suspicious similarities between changes in 10Be concentrations and proxy time-series in TSK and JC sediments. The resulting significant contributions to the multi-regressions with 10Be by TOC and Ca for both TSK and JC point to an influence of these proxies on our 10Be records. Therefore, we discuss in a second step the likely mechanisms behind the statistical linkages based on existing literature. This discussion supports the statistically inferred results, pointing to a preferential binding of 10Be to organic material and a reduced affinity for Ca (we provide additional references to support these findings). Third, based on the performed statistical analyses and inferred chemical behavior, we correct our 10Be concentration time-series from TSK and JC by subtracting the calculated bias in 10Be imprinted by TOC and Ca variations. The above analyses and corrections are exclusively based on the statistic similarities between the 10Be and TOC/Ca proxy records as well as the inferred chemical behavior of 10Be in lake sediments. They are not guided by similarities between our corrected 10Be time-series and 14C production rate variations and, hence, not subject to circle reasoning. Since we could not detect significant contributions of Ti, Fe and SAR to the multi-regressions, we did not include these proxies to the correction procedure. Comparable linkages between 10Be and TOC, Ca, Ti, Fe and SAR were found in a previous study on 10Be in sediments from Lake Meerfelder Maar, supporting our findings (Czymzik et al., 2016, Quaternary Science Reviews).

The extended and restructured Chapter 5.1.: Environment and catchment conditions can add non-production variations to 10Becon records from lake sediments (Berggren et al., 2010; Czymzik et al., 2015). To detect and reduce these variations, we perform

a three-step statistical procedure following Czymzik et al. (2016a), with a slight modification. First, multi-linear regressions were calculated between the 10Becon records and TOC, SAR, Ca, Si, Ti proxy time-series from TSK and JC, reflecting changes in sediment accumulation and composition (Dräger et al., 2017; Ott et al., 2016; Wulf et al., 2016), to estimate the possible environmental influence (10Bebias) (Figs. 2, 3 and S1). Only the TOC and Ca time-series significantly contributed (p<0.1) to the final multi-regressions with 10Be for both TSK and JC sediments. These statistical connections between 10Be and TOC / Ca for TSK and JC point to depositional mechanisms of 10Be in lake sediment archives. Significant contributions to the multi-regression as well as significant positive correlations for TSK (r=0.62, p<0.01) and JC (r=0.77, p<0.01) suggest a preferential binding of 10Be to organic material (Figs. 2 and 3). This result is supported by significant positive correlations of 10Be with TOC in two annually resolved time-series from varved sediments of TSK and JC spanning solar cycles 22 and 23 and in Meerfelder Maar sediments covering the Lateglacial-Holocene transition (Czymzik et al., 2015, 2016a). Significant contributions of Ca to the multi-regressions as well as significant negative correlations with 10Be for TSK (r=-0.68, p<0.01) and JC (r=-0.62, p<0.01) might point to a reduced affinity of 10Be for Ca (Figs. 2 and 3). A similar behavior was detected in studies about 10Be scavenging from the marine realm (Aldahan and Possnert, 1998; Chase et al., 2002, Simon et al., 2016). Based on the statistical connections and inferred chemical behavior of 10Be in sediments, the 10Bebias time-series from TSK and JC sediments were subtracted from the original 10Becon records in an attempt to construct an environment-corrected version of the 10Be record (10Beenvironment). However, this statistical approach also removes variability in the 10Becon records only coincident with variations in proxy time-series, but without a mechanistic linkage, potentially resulting in an overcorrection. Such coinciding variability can, for example, be introduced by solar activity variations causing 10Be production rate changes and climate variations imprinted in the proxy time-series. Therefore, final 10Be composite records (10Becomp) were calculated by averaging the 10Becon and 10Beenvironment records from each site. To enhance the robustness of

the corrections, the procedure was performed on the complete 10Becon records from TSK and JC covering all three grand solar minima. Uncertainty ranges of the calculated 10Becomp records are expressed as the differences between the 10Becon and 10Beenvironment time-series (Fig. 4). Calculated 10Becomp time-series from TSK and JC sediments yield modified trends, but similar multi-decadal variability as the original 10Becon records during the Maunder- (TSK: r=0.84, p<0.01; JC: r=0.91; p<0.01), Homeric- (TSK: r=0.81, p<0.01; JC: r=0.74; p<0.01) and 5500 a BP grand solar minima (TSK: r=0.89, p<0.01; JC: r=0.68; p<0.01) (Fig. 4). These linkages suggest that our correction procedure predominantly reduced trends in the 10Becon records introduced by varying sedimentary TOC and Ca contents, but largely preserved multi-decadal variations connected with varying 10Be production rates (Figs. 2, 3 and 4). Comparable linkages between measured and corrected 10Be records (based on a similar approach) were found in Lake Meerfelder Maar sediments covering the Lateglacial-Holocene transition as well as in recent TSK and JC sediments (Czymzik et al., 2015, 2016a).

(2) Original TSK and JC chronologies

We agree with the reviewer about the importance of the original TSK and JC chronologies for our synchronization study. Therefore, we added the new Chapter 3.3 'Chronologies' to the methods section of the manuscript providing an overview on that subject. For published details on the original TSK and JC chronologies we refer to the related papers by Dräger et al. (2016, The Holocene), Ott et al. (2016, Journal of Quaternary Science; 2017, The Holocene) and Wulf et al. (2013, Quaternary Science Reviews). In addition, we now report the uncertainties of the original TSK and JC chronologies during the investigated three grand solar minima in the new Chapter 3.3.

The new Chapter 3.3. 'Chronologies': The age models for TSK and JC sediments were constructed using a multiple-dating approach. Microscopic varve counts were carried out for both lake sediments. Non-varved intervals in TSK sediments were bridged based on varved thickness measurements in neighbouring well-varved sediment sections. Independent age control for the TSK and JC varve chronologies was provided

by radiocarbon dating and tephrochronology (for details see: Dräger et al., 2016, Ott et al., 2016, 2017; Wulf et al., 2013). Resulting chronological uncertainties are ± 17 (TSK) and ± 4 years (JC) for the Maunder Minimum, ± 139 (TSK) and ± 29 years (JC) for the Homeric Minimum as well as ± 74 (TSK) and ± 56 years (JC) for the 5500 a BP grand solar minimum (see Fig. 6).

Point-by-point reply to reviewer Quentin Simon:

(1) 2.5. The authors could explicitly mention all type of archives which (will) benefit from 10Be for global synchronization. What is the range of time-scale uncertainties associated with these different archives? This is particularly important since it implies different resolutions associated with inherent archive limitations. Despite the most robust archive-to-archive correlation possible (maybe provided by 10Be), these restrictions constitute a limiting factor for studying specific climatic mechanisms in some archives and/or from older ages, particularly about precise lead and lags in the climate system. We wrote in the previous version of our manuscript (lines 2.25) that, to date, mainly ice core and tree archives benefit from the cosmogenic radionuclide synchronization method.

Following the reviewer, we now further specify in line 2.30 that in general sedimentary archives (marine and terrestrial) could profit from our new approach and added that the possible temporal resolution of a synchronization study is limited mainly by the lowest resolution of the involved records.

(2) 2.10. The authors can add paleomagnetism to the series of useful synchronization tools independent from climatic cycles

We added paleomagnetism to the list of synchronization tools and provided the reference Stanton et al. (2010, Quaternary Geochronology) synchronizing a lake sediment record from Sweden to paleointensity variations.

(3) 2.15. Recent works of groups from, e.g., France (Ménabréaz, Valet, Simon) or

Japan (Suganuma, Horiuchi) also documented geomagnetic field forcing on the 10Be production variation, is there an impact of these modulation on your records? More largely, what is the impact of solar activity and geomagnetic intensity variations on the magnitude of atmospheric 10Be production rates? Since authors are discussing a synchronization tool that can (will) be used for other time periods, presenting these elements is important because they explain why and how 10Be works, particularly at certain period of time.

We have to be more distinct about the role of both changes in solar activity (mainly on < 500-year time-scales) and paleomagnetic field intensity (mainly on > 500-year time-scales) on 10Be production rate changes. Therefore, in addition to the reference to Snowball and Muscheler (2007, The Holocene), we now emphasize the different time-scales connected with 10Be production rate changes induced by solar activity variations (decadal to centennial) and geomagnetic field strength (sub-millennial and longer) by providing the additional references Stuiver and Braziunas (1989, Nature) and Simon et al. (2016, Journal of Geophysical Research).

(4) 2.25. What is the result of this spatial heterogeneity? Does it complicate easy interregional correlations? If yes, to what extent? This is important for using 10Be as an accurate global synchronization tool of course.

Spatially inhomogeneous deposition is one of the major uncertainties in 10Be research. However, due to the same production mechanism and different geochemical behavior, the shared variance of 10Be and 14C records can be considered to reflect common atmospheric cosmogenic radionuclide production variations (e.g. Muscheler et al., 2014, Quaternary Science Reviews). That is one of the reasons why we compare our 10Be records from TSK and JC sediments to 14C production variations from the tree-ring based part of the IntCal13 calibration curve. We added that information in lines 2.25.

(5) 2.30. Add the recent Raisbeck et al. paper (Clim. Past, 13, 217–229, 2017) which discusses synchronization between Greenland and Antarctic ice cores using 10Be.

Does "synchronization of terrestrial paleoclimate records around the globe" need to assume a global homogenization of 10Be production/deposition (see above)?

We now provide the reference Raisbeck et al. (2017, Climate of the Past). See our answer to comment 4, dealing with spatial inhomogeneous 10Be deposition.

(6) 2.35. Why only studying these three periods? Do you expect higher level of 10Be changes during these intervals? It might be interesting to give some precision here. Moreover, it could be a good idea to mention the three grand solar minima it in the title since your study is focused on these periods.

We added the 'three grand solar minima' to the title. In lines 3.5 we specify that the three grand solar minima were chosen, because they comprise among the lowest solar activity levels during the last 6000 years and provided the reference to the solar activity reconstruction by Steinhilber et al. (2012, PNAS).

(7) 3.15. What is the extent of sedimentary changes in both cores through the studied intervals? Are they related to any known (studied) climatic cycle? This is important since sedimentary changes can drastically disturb Be records in geological archives. For instance, the last two sentences dealing with current air masses and precipitations behavior are interesting for modern settings but do these parameters also prevailed during the periods scrutinized here?

See our detailed answer 1 'Changes in sediment composition and their effects on 10Be deposition'.

(8) 3.20. To what range of depth intervals correspond a 20-year resolution? What is the sediment amount needed for method? How many years are integrated by the sampling (thickness of the sediment samples)? Also, I do understand that authors want to keep short, which is definitely not a bad idea, but since the chronology is central in the paper (e.g. Title) I find important to present how age models have been obtained (not simply referring to the original publications). What is their resolutions

and uncertainties? There is no need to develop too far, but to provide with enough elements for the readers to judge the resolution and potential bias induced by inevitable age errors. This is particularly important since the paper discusses about age offsets with resolutions of only few years back to > 5 ka BP.

See our detailed answer 2 'Chronology'. We added to Section 3.1. 'Sediment sub-sampling and proxy records' that a 20-year resolution equals on average about 20 mm sediment.

(9) 3.30/4.5. How do you homogenize sediment samples? What is the sediment weight used? Authors should write that they are interested only by the fraction adsorb or precipitated on sediments (sometimes called "authigenic"), and why are they interested by this fraction? They could precise that metal hydroxides and silicates are precipitated while Be remains in solution. Why precipitate at pH 10 and not 8.5? Are you not precipitating (or risk to precipitate) Boron at this pH level? These last two questions are probably not interesting for the paper, personal interest about the method. It could be useful to add a citation that provide with full description of the method followed here. Add at the end of the last sentence: "and corrected for radioactive decay (Chmeleff et al., 2010; Korschinek et al., 2010)". I totally understand that this correction does not change your results, but better be precise with radioactive elements. Retrieving the exact 10Be concentrations imply a correction for radioactive decay, even if changes occur at the margin given the sediment ages and the T1/2 of 10Be. Note also that authors could add somewhere in the text the half-life of both 14C and 10Be to give the time extent, and therefore theoretical limits, of these tracers (probably more useful for 10Be than for 14C which is already well known by the community).

We added the methodological information requested by Quentin Simon to the text. Moreover, we now provide information about the half-time of 10Be and the effects of radioactive decay on our time-series in the 'Results' section by adding the sentence: 'Due to the 1.387 $\pm$ 0.012 Ma long half-life of 10Be (Korschinek et al., 2010) the effect of radioactive decay is negligible in our 10Be records'.

(10) 4.10/15. What is the time uncertainty associated with your data?

See our detailed answer 2 'Chronology'.

(11) 4.20. I would remove any mention to Figs. 2 and 3 in the results section as these figures are plotted versus age. Results versus ages are already part of a discussion because they imply a serious transformation through the application of age modeling. Presentation of the raw 10Be concentration data versus depth in new figures is maybe not mandatory since I guess these data will be available as supplementary material or easily available from the web. I know this comment is annoying but discussion will likely evolve while the data will remain, and are therefore important for the community. The authors should highlight directly on the figures the location and duration interval of the grand solar minima discussed (which do not represent the whole box intervals).

We added Figure S1 to the manuscript depicting a plot of our new 10Be data from TSK and JC against sediment core depth. We now highlight the investigated three grand solar minima in Fig. 6 using arrows.

(12) 4.30. What kind of non-production forcing parameters can explain part of the 10Be concentration variations in varved lake sediments? 2-3 sentences could help readers to rapidly understand such processes without having to refer to a third party (interested readers will of course go to these citations). I'm wondering why you selected these parameters specifically (i.e. TOC, SAR, Ca, Si, Ti)? Are their fluctuations representing correctly all lithological changes observed in the lakes (e.g. productivity, grain-size, mineralogy)?

See our detailed answer 1 'Changes in sediment composition and their effects on 10Be deposition'.

(13) 5.5. I agree that significant contribution of TOC and Ca on the whole intervals justify their used to the multi-regressions treatment. Yet, it is possible that other elements presented in the paper also impact the 10Be signal within specific depth intervals (e.g.

[Figure]

Ti since about 200 a BP in TSK). If they are associated with specific events linked to rapid climatic changes, how can you estimate their residual influence on the 10Beenvironment record calculated? Actually authors correctly discuss that matter later in the paragraph but it results into a blurry questioning about the reliability of the environmental correction procedure, essentially because the method does not rely on any mechanistic linkages between 10Beconc and TOC/Ca, as mentioned by the authors themselves. One could mention here that the method is mainly working because the outcome (10Becomp) is highly comparable with 14C production (Fig. 6) but, although valid, this argument is slightly circular. The main question remains: how to correctly remove, or say diminish, environmental variability imprints on 10Be records in lakes?

See our detailed answer 1 'Changes in sediment composition and their effects on 10Be deposition'.

(14) 5.15. As the authors are interested by multi-decadal variations (see Figures 5 and 6), why not working on 10Beconc series directly as this variability is similar between both 10Becomp and 10Beconc series. This would avoid unnecessary and questionable data treatments while preserving the conclusion.

See our detailed answer 1 'Changes in sediment composition and their effects on 10Be deposition'.

(15) 5.25/25. These two paragraphs are rather interesting but could be move above (5.5) to support the use of these two elements for the multi-regression method used to obtain the 10Bebias. Also, it would be interesting to discuss a little bit more (or cite references?) about the exact – or supposed – mechanisms explaining "preferential binding of 10Be to organic material", while the affinity of 10Be to Ca has been indeed demonstrated in several studies already cited in the paper.

Please see our detailed answer 1 'Changes in sediment composition and their effects on 10Be deposition'.

(16) 6.5. Does result differs when using 10Beconc instead of 10Becomp (see comment above)? In TSK, the unfiltered two 10Be peaks visually correlated with the two sunspot number lows, why not mentioning it? Do you have sedimentological elements to sustain a transport of "old" 10Be? By which processes such a transport can take place (physical remobilization or desorption form sediments previously deposited onto "shelves")? It might be interesting to mention it here, or to refer to explanations provide later in the text.

See our detailed answer 1 'Changes in sediment composition and their effects on 10Be deposition'.

(17) 6.20. Are you using this best fit result to propose a new chronology for TSK?

Yes, the successful synchronizations will be used as tie-points to improve the chronologies of TSK and JC. We added 'improved chronologies' to the conclusions. The extended sentence from the conclusions: 'These synchronizations provide a novel type of time-marker for varved lake sediment archives enabling improved chronologies and robust investigations of proxy responses to climate variations'.

(18) 7.5/10. Conclusion is fine and clearly wrap up the main objective of the paper, i.e. 10Be is a robust tool for synchronization (TSK) unless environmental imprint is too strong (JC).

Thank you!
* * *
[Figure]

**Fig. 1.** Supp. Fig. 1

---

## Referee Comment (RC2) · Anonymous Referee #2 · 13 Mar 2018

General comments

This manuscript by Czymzik and co-authors targets to a key issue in paleoclimate records i.e. time-scale uncertainties, which often inhibit the detailed investigation of multiple spatial high resolution climate proxy records. 10Be records from two varved lake sediment sequences from northern Germany and Poland are synchronized with IntCal13 calibration curve. This methodological approach is a novel attempt to synchronize lake sediment records using 10Be in order to investigate the leads and lags, unwanted but inherent features in all proxy records. Large (and growing) number of the high resolution paleoclimatic studies is published from lacustrine sediments but the detailed comparison of the proxy records suffer from the temporal uncertainties. From this perspective, the manuscript contains interesting ideas and is topical. The text is

well written and structured and has illustrations of high quality to support results and interpretations very nicely. The main point what I miss in this manuscript would be a visual illustration of the sediment composition and composition changes from the two sediment records with SAR, TOC and perhaps Ca, Ti and 10Be variability, at least for the time windows that were more closely inspected. Although the references to original publications are provided, the illustration would greatly help to follow the detailed discussion from two lake records with several proxies and time windows and changes in sedimentation. Overall, this manuscript is suited for the journal of Climate of the Past discussions and can be accepted with minor revision.

Specific comments

Page 2 Line 26: Could it be shortly explained how the non-uniform 10Be depositional patterns are generally taken into account/expected to influence the records?

Page 3 Line 9: No major inflows, today. Well, were there major inflows previously? What kind of changes in inflow system have occurred and when? Does this influence the sediment composition within the time interval of the study, e.g. the changes in sedimentation rate or sediment composition? If not, this should be mentioned as well.

Page 3 Line 20: at 20 year resolution. This is not clear to me; do you mean one sample every 20 years, or a sample comprising 20 years?

Page 3 Methods – Page 4 Results: Overall, this section leaves me a bit confused. For a reader I feel I am left with a tenuous grasp on the TSI and CJ records. Although the references are provided it would be helpful to shed light on these previously published varve records that are frequently referred in the text, e.g. where the non-varved sections are located and how the sediment composition changes (in time/depth scale)? An illustration of the records perhaps with some Ti, Ca TOC and even 10Be variation curves would be helpful to quickly get an overall picture of the two records.

Page 4 Line 27: Although references are provided it would be helpful to mention briefly

how environmental and catchment conditions can influence the 10Becon variation in sediment record.

Page 5 Line 12: The correlations could be added in the figure 4 similarly as is done in figures 2 and 3.

Page 5 Line 19: Could these depositional mechanisms be briefly described?

Page 5 Line 20-21: At this point it does not become clear which correlations are referred. This becomes clear later in the paragraph but text would be easier to follow if the correlations were specified before showing the numbers.

Page 6 Line 8-9: Why? Are there indications in the sediments that suggest resuspension of littoral sediments or changes in sediment focusing? The illustration of sediment composition (see general comments) could be helpful here.

Page 6 Line 29-30: This (also) would be nicely clarified with the record-describing illustration (see comment for Page 3-4 Methods-Results).

Page 7 Line 1-2: What is this layer? Does this occur at the time interval discussed in this paper at Page 6 Line 6 (from -50 to 0 BP)? If so, this could be mentioned already earlier. This would actually answer partly to the specific comment I made for Page 6 Line 8-9.

Figure 4: Why 10Becon and 10Beenvironment are out of phase in Lake Czechowskie from about 2700 to 3100 BP?

---

## Author Comment (AC2) · 22 Mar 2018

We thank the Anonymous Referee #2 for his thorough comments, which helped to significantly improve the manuscript. In the following, we give a detailed response to all concerns raised, first answering the main point of criticism, followed by a point-by-point reply.

General comments:

This manuscript by Czymzik and co-authors targets to a key issue in paleoclimate records i.e. time-scale uncertainties, which often inhibit the detailed investigation of multiple spatial high resolution climate proxy records. 10Be records from two varved lake sediment sequences from northern Germany and Poland are synchronized with

IntCal13 calibration curve. This methodological approach is a novel attempt to synchronize lake sediment records using 10Be in order to investigate the leads and lags, unwanted but inherent features in all proxy records. Large (and growing) number of the high resolution paleoclimatic studies is published from lacustrine sediments but the detailed comparison of the proxy records suffer from the temporal uncertainties. From this perspective, the manuscript contains interesting ideas and is topical. The text is well written and structured and has illustrations of high quality to support results and interpretations very nicely.

The main point what I miss in this manuscript would be a visual illustration of the sediment composition and composition changes from the two sediment records with SAR, TOC and perhaps Ca, Ti and 10Be variability, at least for the time windows that were more closely inspected. Although the references to original publications are provided, the illustration would greatly help to follow the detailed discussion from two lake records with several proxies and time windows and changes in sedimentation. Overall, this manuscript is suited for the journal of Climate of the Past discussions and can be accepted with minor revision.

Detailed Answer: We visualize changes in sediment composition and accumulation by depicting the original 10Be, TOC, Si, Ca, Ti and SAR records from Lakes Tiefer See and Czechowskie for the inspected three grand solar minima in Figs. 2 (for TSK) and 3 (for JC). In addition, we added the supplementary Fig. S1 to the revised manuscript showing our new 10Be records from Lakes Tiefer See and Czechowskie against core depth.

Specific comments:

Page 2 Line 26: Could it be shortly explained how the non-uniform 10Be depositional patterns are generally taken into account/expected to influence the records?

Non-uniform deposition patterns are presently one of the main uncertainties in 10Be research (Adolphi and Muscheler, 2016, Climate of the Past). However, common
changes of the cosmogenic radionuclides 10Be and 14C in different environmental archives are considered to reflect variations in the cosmogenic radionuclide production rate, due to their same production mechanism and different chemical behavior (Muscheler et al., 2016, Solar Physics). That is one of the reasons why we compare our 10Be time-series from Lakes Tiefer See and Czechowskie to 14C production rates inferred from the IntCal13 calibration curve. To account for the reviewer's comment, we added the following sentence in page 2 lines 27-29 of the revision and provide two references:

'Despite these non-production effects, common changes in 10Be and 14C records are considered to reflect the cosmogenic radionuclide production signal, due to their common production mechanism and different chemical behavior (Lal and Peters, 1967, Muscheler et al., 2016).'

Another way to distinguish and reduce non-production effects in sedimentary 10Be time-series is our here applied approach based on environmental proxy-series from the same archive. Thereby, it is assumed that coinciding changes the environment reflected by proxy time-series might leave an imprint in the 10Be time-series (Czymzik et al., 2016, Quaternary Science Reviews).

Page 3 Line 9: No major inflows, today. Well, were there major inflows previously? What kind of changes in inflow system have occurred and when? Does this influence the sediment composition within the time interval of the study, e.g. the changes in sedimentation rate or sediment composition? If not, this should be mentioned as well.

Very low and rather stable contents of detrital grains in varved Lakes Tiefer See and Czechowskie sediments indicate that no major tributaries existed throughout the investigated three grand solar minima (and the complete Holocene). To include this information to the manuscript we revised the related sentence and provide two references.

'The lake basins are part of subglacial channel systems formed at the end of the last glaciation and had no major inflows during the Holocene (Dräger et al., 2017; Ott et al.,

2016).'

Page 3 Line 20: at 20 year resolution. This is not clear to me; do you mean one sample every 20 years, or a sample comprising 20 years?

We need to be clearer about our sampling strategy. We use continuous series of sediment samples, each comprising about 20 years of sedimentation. Therefore, we changed the related sentence to:

'Continuous series of sediment samples at 20-year resolution were extracted for 10Be measurements from sediment cores TSK11 and JC-20 M2015, based on varve chronologies (Dräger et al., 2017; Ott et al., 2016).'

Page 3 Methods – Page 4 Results: Overall, this section leaves me a bit confused. For a reader I feel I am left with a tenuous grasp on the TSI and CJ records. Although the references are provided it would be helpful to shed light on these previously published varve records that are frequently referred in the text, e.g. where the non-varved sections are located and how the sediment composition changes (in time/depth scale)? An illustration of the records perhaps with some Ti, Ca TOC and even 10Be variation curves would be helpful to quickly get an overall picture of the two records.

See our Detailed Answer on Figs. 2 and 3 showing the original 10Be, Ti, Ca, Si, TOC and SAR data from Lakes Tiefer See and Czechowskie sediments during the investigated three grand solar minima. Moreover, we now add information about non-varved sediment sections in Lake Tiefer See to Fig. 2, by highlighting the respective time-intervals with bars. We describe this new feature in the caption to Fig. 2.

Page 4 Line 27: Although references are provided it would be helpful to mention briefly how environmental and catchment conditions can influence the 10Becon variation in sediment record.

We discuss possible mechanistic linkages between environmental effects and 10Be deposition in Lakes Tiefer See and Czechowskie sediments in detail at the end of

Chapter 5.1, after describing and performing our procedure used for the attribution and correction of these effects. To be clearer about this structure, we added a sentence and reformulated the beginning of Chapter 5.1:

'Environment and catchment conditions can add non-production variations to 10Becon records from varved lake sediments (Berggren et al., 2010; Czymzik et al., 2015). In the following we will, first, describe and perform our approach used for detecting and correcting for possible non-production features in our 10Be time-series and, then, discuss possible mechanisms behind the statistically inferred connections.'

Page 5 Line 12: The correlations could be added in the figure 4 similarly as is done in figures 2 and 3.

Done.

Page 5 Line 19: Could these depositional mechanisms be briefly described?

See our answer to comment 'Page 4, Line 27'.

Page 5 Line 20-21: At this point it does not become clear which correlations are referred. This becomes clear later in the paragraph but text would be easier to follow if the correlations were specified before showing the numbers.

We now mention the correlation between 10Be and TOC before we provide the correlation coefficient and significance level.

Page 6 Line 8-9: Why? Are there indications in the sediments that suggest resuspension of littoral sediments or changes in sediment focusing? The illustration of sediment composition (see general comments) could be helpful here.

See our Detailed Answer. We added more detailed information on the sub-layer of resuspended littoral sediments in JC varves, deposited back to 2800 a BP, in Section 5.3. However, to avoid repetition, we prefer at this point of the text not to go into details and hint more clearly to the later discussion:

'In JC sediments, the 10Becomp excursion from -50 to 0 a BP (AD 2000-1950) without an expression in the group sunspot number record as well as the about 20-year delayed Maunder Minimum response could be explained by transport of 'old' 10Be from the littoral to the coring site (see more details on the sub-layer of resuspended littoral sediments in JC varves back to 2800 a BP in Section 5.3) and/or spatially inhomogeneous 10Be deposition patterns (Fig. 5).'

Page 6 Line 29-30: This (also) would be nicely clarified with the record-describing illustration (see comment for Page 3-4 Methods-Results).

See our Detailed Answer.

Page 7 Line 1-2: What is this layer? Does this occur at the time interval discussed in this paper at Page 6 Line 6 (from -50 to 0 BP)? If so, this could be mentioned already earlier. This would actually answer partly to the specific comment I made for Page 6 Line 8-9.

This sub-layer was deposited in fall AD 2003. It consists of the same littoral diatoms and patches of calcite deposited in Lake Czechowskie during this season since about 2800 a BP. However, during that year the fall layer was exceptionally thick (3.7 mm), containing comparably high amounts of 'old' 10Be leading to an anomalous 10Be value for that year (Czymzik et al., 2015, Earth and Planetary Science Letters). To account for the reviewer's comment, we added more information about this exceptional sediment sub-layer to the manuscript.

'Comparable influences of sediment resuspension were also found in a sample from an annually resolved 10Be time-series of recent JC sediments covering the period AD 2009-1988 (Czymzik et al., 2015). A varve with an exceptionally thick (3.7 mm) layer of resuspended littoral diatoms and patches of calcite deposited in fall 2003 reveals anomalous 10Be values (Czymzik et al., 2015).'

Figure 4: Why 10Becon and 10Beenvironment are out of phase in Lake Czechowskie

from about 2700 to 3100 BP?

This is an effect of our 'environmental correction' procedure. When the correction is large, the generated signal will look increasingly different from the original 10Becon record. That this looks partly like a phase shift is mere coincidence. We discuss on page 6, lines 11-20, that we do not obtain significant fits between Lake Czechowskie 10Be and IntCal13 14C production during the Homeric Minimum and point out that this is likely due to environmental influences on Lake Czechowskie 10Be, which are challenging to correct for.

Thank you!

---

## Author Response (AR1)

[revised manuscript text omitted]

**Response to the reviewers' comments**

We thank Quentin Simon and an anonymous reviewer for their constructive and detailed comments which helped to significantly improve our manuscript. In the following, we will give a detailed response to all concerns that have been raised, first answering the main points of criticism, followed by a point-by-point reply.

**Response to reviewer Quentin Simon:**

**Detailed Answer #1: Changes in sediment composition, their effects on $^{10}$Be deposition and how to correct for them**

Apparently, we were not clear enough about the possible effects of changes in sediment composition on $^{10}$Be deposition in TSK and JC, and how we try to reduce them. Therefore, we modified our manuscript in two ways:

  (1) To improve the overview on varying $^{10}$Be concentrations and sediment composition in TSK and JC, we now show in addition to Figs. 2 and 3 ($^{10}$Be concentrations, Ti, SAR, TOC, Si and Ca proxy time-series from TSK/JC vs. time) our $^{10}$Be records against sediment core depth in the new Figure S1, as suggested by Quentin Simon. Our $^{10}$Be data on age and depth scale will be made available to the public in the PANGAEA data library.

  (2) To be clearer about our research strategy, we have extended the introductory paragraph of Chapter 5.1. There, we now outline that we first present our statistical approach (multi-regression analyses) applied to detect suspicious similarities between changes in $^{10}$Be concentrations and proxy time-series in TSK and JC sediments and, second, discuss possible mechanistic linkages behind the inferred connections.

  The revised and extended introduction to Chapter 5.1. (lines 5.11-14):

  Environment and catchment conditions can add non-production variations to $^{10}$Be$_{con}$ records from varved lake sediments (Berggren et al., 2010; Czymzik et al., 2015). In the following chapter we will, first, describe our approach used for detecting and correcting possible non-production features in our $^{10}$Be time-series and, then, discuss possible mechanisms behind the statistically inferred connections.

Moreover, the above detection and correction procedures are exclusively based on statistic similarities between the $^{10}$Be and other proxy records from the same archive as well as the inferred chemical behavior of [10]Be in lake sediments. They are not guided by similarities
between our corrected [10]Be time-series and [14]C production rate variations and, hence, not
subject to circle reasoning. Since we could not detect significant contributions of Ti, Fe and SAR
to the multi-regressions, we did not include these proxies to the correction procedure.
Supporting our findings, similar linkages between [10]Be and TOC, Ca, Ti, Fe and SAR were
found in a previous study on [10]Be in sediments from Lake Meerfelder Maar (Czymzik et al.,
2016, *Quaternary Science Reviews*).

**42 Detailed answer #2: Original chronologies**

We agree with the reviewer about the importance of the original TSK and JC chronologies for
our synchronization study. Therefore, we added the new Chapter 3.3 'Original chronologies' to
the methods section of the manuscript providing an overview on that subject. For published
details on the original TSK and JC chronologies we refer to the related papers by Dräger et al.
(2016, *The Holocene*), Ott et al. (2016, *Journal of Quaternary Science*; 2017, *The Holocene*)
and Wulf et al. (2013, *Quaternary Science Reviews*). In addition, we now report the
uncertainties of the original TSK and JC chronologies during the investigated three grand solar
minima.

The new Chapter 3.3. 'Original chronologies':

The age models for TSK and JC sediments were constructed using a multiple-dating approach.

Microscopic varve counts were carried out for both lake sediments. Non-varved intervals in TSK

sediments were bridged based on varved thickness measurements in neighbouring well-varved sediment sections. Independent age control for the TSK and JC varve chronologies was provided by radiocarbon dating and tephrochronology (for details see: Dräger et al., 2016, Ott et al., 2016, 2017; Wulf et al., 2013). Resulting chronological uncertainties are ± 17 (TSK) and ± 4

years (JC) for the Maunder Minimum, ± 139 (TSK) and ± 29 years (JC) for the Homeric

Minimum as well as ± 74 (TSK) and ± 56 years (JC) for the 5500 a BP grand solar minimum (see Fig. 6).

**63 Point-by-point reply to reviewer Quentin Simon:**

In this paper, the authors address an essential issue in any paleo-studies, the chronology. They
propose a method to help synchronizing varved lakes with other natural archives which is
essential to improve our understanding of the mechanisms driving climatic variability. The group
of authors already introduced this method for other sites or other time periods in several
publications and pursued successfully their investigations in this paper that I recommend for publication following some revisions. The aim of the paper is essentially methodological and I imagine that climatic discussion based on precise inter-correlation of TSK (less evident for JC) with other records, using the 10Be method, will be presented elsewhere (which is fair). The text would greatly benefit from several complementary notes on method, interpretation and discussion (see comment below). Given the rather short length of the present manuscript, it should be possible to provide these information without weighing too much on the final version of the paper. I listed below a series of comments and questions – voluntarily naïve or not – for which answers (integrated into the paper) should improved robustness an easy readability of the paper.

I would also suggest a slight change of the title to enclose all aspects of the paper: "Synchronizing 10Be in two varved lake sediment records to IntCal13 14C during grand solar minima" (see below).

**(1)** 2.5. The authors could explicitly mention all type of archives which (will) benefit from 10Be for global synchronization. What is the range of time-scale uncertainties associated with these different archives? This is particularly important since it implies different resolutions associated with inherent archive limitations. Despite the most robust archive-to-archive correlation possible (maybe provided by 10Be), these restrictions constitute a limiting factor for studying specific climatic mechanisms in some archives and/or from older ages, particularly about precise lead and lags in the climate system.

> In the previous version of our manuscript we wrote that, to date, mainly ice core and tree archives benefit from the cosmogenic radionuclide synchronization method. Following the reviewer, we now further specify in lines 2.33-34 that in general sedimentary archives (marine and terrestrial) could profit from our new approach and added that the possible temporal resolution of a synchronization study is limited mainly by the lowest resolution of the involved records.

**(2)** 2.10. The authors can add paleomagnetism to the series of useful synchronization tools independent from climatic cycles

> We added paleomagnetism to the list of synchronization tools and provided the reference Stanton et al. (2010, *Quaternary Geochronology*). This study synchronizes a varved lake sediment record from Sweden based on paleointensity variations.

**(3)** 2.15. Recent works of groups from, e.g., France (Ménabréaz, Valet, Simon) or Japan (Suganuma, Horiuchi) also documented geomagnetic field forcing on the 10Be production variation, is there an impact of these modulation on your records? More largely, what is the impact of solar activity and geomagnetic intensity variations on the magnitude of atmospheric 10Be production rates? Since authors are discussing a synchronization tool that can (will) be used for other time periods, presenting these elements is important because they explain why and how 10Be works, particularly at certain period of time.

We have to be more distinct about the role of both changes in solar activity (decadal to
centennial) and paleomagnetic field intensity (sub-millennial and longer) on $^{10}$Be
production rate changes. Therefore, in addition to the reference to Snowball and
Muscheler (2007, *The Holocene*), we now emphasize the different time-scales
connected with $^{10}$Be production rate changes induced by solar activity variations and
geomagnetic field strength by providing the additional references Stuiver and Braziunas
(1989, *Nature)* and Simon et al. (2016, *Journal of Geophysical Research*).

**(4)** 2.25. What is the result of this spatial heterogeneity? Does it complicate easy interregional
correlations? If yes, to what extent? This is important for using 10Be as an accurate global
synchronization tool of course.
Spatially inhomogeneous deposition is one of the major uncertainties in $^{10}$Be research.
However, due to the same production mechanism and different geochemical behavior,
the shared variance of $^{10}$Be and $^{14}$C records are considered to reflect common
atmospheric cosmogenic radionuclide production variations (e.g. Muscheler et al., 2014,
*Quaternary Science Reviews*). That is one of the reasons why we compare our $^{10}$Be
records from TSK and JC sediments to $^{14}$C production variations from the tree-ring
based part of the IntCal13 calibration curve. We added that information in lines 2.27-29.

**(5)** 2.30. Add the recent Raisbeck et al. paper (Clim. Past, 13, 217–229, 2017) which discusses
synchronization between Greenland and Antarctic ice cores using 10Be. Does "synchronization
of terrestrial paleoclimate records around the globe" need to assume a global homogenization of
10Be production/deposition (see above)?
We now provide the reference Raisbeck et al. (2017, *Climate of the Past*). See our
answer to Comment 4, dealing with spatial inhomogeneous $^{10}$Be deposition.

**(6)** 2.35. Why only studying these three periods? Do you expect higher level of 10Be changes
during these intervals? It might be interesting to give some precision here. Moreover, it could be
a good idea to mention the three grand solar minima it in the title since your study is focused on
these periods.
We added the 'three grand solar minima' to the title. In lines 3.7-8 we specify that the
three grand solar minima were chosen, because they comprise among the lowest solar
activity levels during the last 6000 years and provided the reference to the solar activity
reconstruction by Steinhilber et al. (2012, *PNAS*).

**(7)** 3.15. What is the extent of sedimentary changes in both cores through the studied intervals?
Are they related to any known (studied) climatic cycle? This is important since sedimentary
changes can drastically disturb Be records in geological archives. For instance, the last two sentences dealing with current air masses and precipitations behavior are interesting for
modern settings but do these parameters also prevailed during the periods scrutinized here?
See our Detailed Answer #1 'Changes in sediment composition and their effects on $^{10}$Be
deposition'.

**(8)** 3.20. To what range of depth intervals correspond a 20-year resolution? What is the
sediment amount needed for method? How many years are integrated by the sampling
(thickness of the sediment samples)? Also, I do understand that authors want to keep short,
which is definitely not a bad idea, but since the chronology is central in the paper (e.g. Title) I
find important to present how age models have been obtained (not simply referring to the
original publications). What is their resolutions and uncertainties? There is no need to develop
too far, but to provide with enough elements for the readers to judge the resolution and potential
bias induced by inevitable age errors. This is particularly important since the paper discusses
about age offsets with resolutions of only few years back to > 5 ka BP.
See our Detailed Answer #2 'Original chronologies'. We added to Section 3.1. 'Sediment
sub-sampling and proxy records' that a 20-year resolution equals on average about 20
mm sediment (lines 3.25-27).
**(9)** 3.30/4.5. How do you homogenize sediment samples? What is the sediment weight used?
Authors should write that they are interested only by the fraction adsorb or precipitated on
sediments (sometimes called "authigenic"), and why are they interested by this fraction? They
could precise that metal hydroxides and silicates are precipitated while Be remains in solution.
Why precipitate at pH 10 and not 8.5? Are you not precipitating (or risk to precipitate) Boron at
this pH level? These last two questions are probably not interesting for the paper, personal
interest about the method. It could be useful to add a citation that provide with full description of
the method followed here. Add at the end of the last sentence: "and corrected for radioactive
decay (Chmeleff et al., 2010; Korschinek et al., 2010)". I totally understand that this correction
does not change your results, but better be precise with radioactive elements. Retrieving the
exact 10Be concentrations imply a correction for radioactive decay, even if changes occur at the
margin given the sediment ages and the T1/2 of 10Be. Note also that authors could add
somewhere in the text the half-life of both 14C and 10Be to give the time extent, and therefore
theoretical limits, of these tracers (probably more useful for 10Be than for 14C which is already
well known by the community).

We added the methodological information requested by Quentin Simon to the text.
Moreover, we now provide information about the half-time of $^{10}$Be and the effects of
radioactive decay on our time-series in the 'Results' section by adding the sentence:

'Due to the 1.387 ± 0.012 Ma long half-life of $^{10}$Be (Korschinek et al., 2010), the effect of
radioactive decay is negligible in our $^{10}$Be records'.

**(10)** 4.10/15. What is the time uncertainty associated with your data?

      See our Detailed Answer #2 'Original chronologies'.

**(11)** 4.20. I would remove any mention to Figs. 2 and 3 in the results section as these figures are plotted versus age. Results versus ages are already part of a discussion because they imply a serious transformation through the application of age modeling. Presentation of the raw 10Be concentration data versus depth in new figures is maybe not mandatory since I guess these data will be available as supplementary material or easily available from the web. I know this comment is annoying but discussion will likely evolve while the data will remain, and are therefore important for the community. The authors should highlight directly on the figures the location and duration interval of the grand solar minima discussed (which do not represent the whole box intervals).

      In addition to Figs. 2 and 3, we now also refer to the new Figure S1 depicting a plot of our new $^{10}$Be data from TSK and JC against sediment core depth. We highlight the three investigated grand solar minima in Fig. 6 using arrows.

**(12)** 4.30. What kind of non-production forcing parameters can explain part of the $^{10}$Be concentration variations in varved lake sediments? 2-3 sentences could help readers to rapidly understand such processes without having to refer to a third party (interested readers will of course go to these citations). I'm wondering why you selected these parameters specifically (i.e. TOC, SAR, Ca, Si, Ti)? Are their fluctuations representing correctly all lithological changes observed in the lakes (e.g. productivity, grain-size, mineralogy)?

      See our Detailed Answer #1 'Changes in sediment composition and their effects on $^{10}$Be deposition'.

**(13)** 5.5. I agree that significant contribution of TOC and Ca on the whole intervals justify their used to the multi-regressions treatment. Yet, it is possible that other elements presented in the paper also impact the 10Be signal within specific depth intervals (e.g. Ti since about 200 a BP in TSK). If they are associated with specific events linked to rapid climatic changes, how can you estimate their residual influence on the 10Beenvironment record calculated? Actually authors correctly discuss that matter later in the paragraph but it results into a blurry questioning about the reliability of the environmental correction procedure, essentially because the method does not rely on any mechanistic linkages between 10Beconc and TOC/Ca, as mentioned by the authors themselves. One could mention here that the method is mainly working because the outcome (10Becomp) is highly comparable with 14C production (Fig. 6) but, although valid, this argument is slightly circular. The main question remains: how to correctly remove, or say diminish, environmental variability imprints on 10Be records in lakes?

See our Detailed Answer #1 'Changes in sediment composition and their effects on $^{10}$Be
deposition'.

**(14)** 5.15. As the authors are interested by multi-decadal variations (see Figures 5 and 6), why
not working on 10Beconc series directly as this variability is similar between both 10Becomp
and 10Beconc series. This would avoid unnecessary and questionable data treatments while
preserving the conclusion.

See our Detailed Answer #1 'Changes in sediment composition and their effects on $^{10}$Be
deposition'.

**(15)** 5.25/25. These two paragraphs are rather interesting but could be move above (5.5) to
support the use of these two elements for the multi-regression method used to obtain the
10Bebias. Also, it would be interesting to discuss a little bit more (or cite references?) about the
exact – or supposed – mechanisms explaining "preferential binding of 10Be to organic material",
while the affinity of 10Be to Ca has been indeed demonstrated in several studies already cited
in the paper.

Please see our Detailed Answer #1 'Changes in sediment composition and their effects
on $^{10}$Be deposition'.

**(16)** 6.5. Does result differs when using 10Beconc instead of 10Becomp (see comment above)?
In TSK, the unfiltered two 10Be peaks visually correlated with the two sunspot number lows,
why not mentioning it? Do you have sedimentological elements to sustain a transport of "old"
10Be? By which processes such a transport can take place (physical remobilization or
desorption form sediments previously deposited onto "shelves")? It might be interesting to
mention it here, or to refer to explanations provide later in the text.

See our Detailed Answer #1 'Changes in sediment composition and their effects on $^{10}$Be
deposition'.

**(17)** 6.20. Are you using this best fit result to propose a new chronology for TSK?

Yes, the successful synchronizations will be used as tie-points to improve the
chronologies of TSK and JC sediments. We added the point 'improving chronologies' to
the conclusions.

The extended sentence from the conclusions (lines 7.30-32):

'These synchronizations provide a novel type of time-marker for varved lake sediment
archives enabling improved chronologies and robust investigations of proxy responses
to climate variations'.

**(18)** 7.5/10. Conclusion is fine and clearly wrap up the main objective of the paper, i.e. 10Be is a
robust tool for synchronization (TSK) unless environmental imprint is too strong (JC).

Thank you!

**Response to Anonymous Referee #2:**
**Detailed answer #3: Visualization of changes in $^{10}$Be and sediment properties:**
We visualize changes in sediment composition and accumulation by depicting the original $^{10}$Be,
TOC, Si, Ca, Ti and SAR records from Lakes Tiefer See and Czechowskie for the inspected
three grand solar minima in Figs. 2 (for TSK) and 3 (for JC). In addition, we added the
supplementary Fig. S1 to the revised manuscript showing our new $^{10}$Be records from Lakes
Tiefer See and Czechowskie against core depth.

**Point-by-point reply to Anonymous Referee #2:**

This manuscript by Czymzik and co-authors targets to a key issue in paleoclimate records i.e.
time-scale uncertainties, which often inhibit the detailed investigation of multiple spatial high
resolution climate proxy records. 10Be records from two varved lake sediment sequences from
northern Germany and Poland are synchronized with IntCal13 calibration curve. This
methodological approach is a novel attempt to synchronize lake sediment records using 10Be in
order to investigate the leads and lags, unwanted but inherent features in all proxy records.
Large (and growing) number of the high resolution paleoclimatic studies is published from
lacustrine sediments but the detailed comparison of the proxy records suffer from the temporal
uncertainties. From this perspective, the manuscript contains interesting ideas and is topical.
The text is well written and structured and has illustrations of high quality to support results and
interpretations very nicely.

The main point what I miss in this manuscript would be a visual illustration of the sediment
composition and composition changes from the two sediment records with SAR, TOC and
perhaps Ca, Ti and 10Be variability, at least for the time windows that were more closely
inspected. Although the references to original publications are provided, the illustration would
greatly help to follow the detailed discussion from two lake records with several proxies and time
windows and changes in sedimentation. Overall, this manuscript is suited for the journal of
Climate of the Past discussions and can be accepted with minor revision.

Specific comments:

Page 2 Line 26: Could it be shortly explained how the non-uniform 10Be depositional patterns
are generally taken into account/expected to influence the records?

Non-uniform deposition patterns are presently one of the main uncertainties in $^{10}$Be
research (Adolphi and Muscheler, 2016, *Climate of the Past*). However, common
changes of the cosmogenic radionuclides $^{10}$Be and $^{14}$C in different environmental
archives are considered to reflect variations in the cosmogenic radionuclide production
rate, due to their same production mechanism and different chemical behavior
(Muscheler et al., 2016, *Solar Physics*). That is one of the reasons why we compare our
$^{10}$Be time-series from Lakes Tiefer See and Czechowskie to $^{14}$C production rates inferred
from the IntCal13 calibration curve. To account for the reviewer's comment, we added
the following sentence to the manuscript and provide two references (lines 2.27-29):

'Despite these non-production effects, common changes in $^{10}$Be and $^{14}$C records are considered to reflect the cosmogenic radionuclide production signal, due to their common production mechanism and different chemical behavior (Lal and Peters, 1967,

Muscheler et al., 2016).'

Another way to distinguish and reduce non-production effects in sedimentary $^{10}$Be time-
series is our here applied approach based on environmental proxy-series from the same
archive. Thereby, it is assumed that coinciding changes the environment reflected by
proxy time-series might leave an imprint in the $^{10}$Be time-series (Czymzik et al., 2016,
*Quaternary Science Reviews*).

Page 3 Line 9: No major inflows, today. Well, were there major inflows previously? What kind of
changes in inflow system have occurred and when? Does this influence the sediment
composition within the time interval of the study, e.g. the changes in sedimentation rate or
sediment composition? If not, this should be mentioned as well.

Very low and rather stable contents of detrital grains in varved Lakes Tiefer See and
Czechowskie sediments indicate that no major tributaries existed throughout the
investigated three grand solar minima (and the complete Holocene). To include this
information to the manuscript we revised the related sentence (lines 3.13-14) and
provide two references:

'The lake basins are part of subglacial channel systems formed at the end of the last glaciation and had no major inflows during the Holocene (Dräger et al., 2017; Ott et al.,

2016).'

Page 3 Line 20: at 20 year resolution. This is not clear to me; do you mean one sample
every 20 years, or a sample comprising 20 years?

We need to be clearer about our sampling strategy. We use continuous series of sediment samples, each comprising about 20 years of sedimentation. Therefore, we changed the related sentence to:

'Continuous series of sediment samples at ~20-year resolution (about 20 mm sediment) were extracted for $^{10}$Be measurements from sediment cores TSK11 and JC-20 M2015, based on varve chronologies (Dräger et al., 2017; Ott et al., 2016).'

Page 3 Methods – Page 4 Results: Overall, this section leaves me a bit confused. For a reader I feel I am left with a tenuous grasp on the TSI and CJ records. Although the references are provided it would be helpful to shed light on these previously published varve records that are frequently referred in the text, e.g. where the non-varved sections are located and how the sediment composition changes (in time/depth scale)? An illustration of the records perhaps with some Ti, Ca TOC and even 10Be variation curves would be helpful to quickly get an overall picture of the two records.

See our Detailed Answer #3 'Visualization of $^{10}$Be and sediment properties'. Moreover, we now add information about non-varved sediment sections in Lake Tiefer See to Fig. 2, by highlighting the respective time-intervals with bars and citing Dräger et al. (2016, *The Holocene*). We describe this new feature in the caption to Fig. 2.

Page 4 Line 27: Although references are provided it would be helpful to mention briefly how environmental and catchment conditions can influence the 10Becon variation in sediment record.

We discuss in detail possible mechanistic linkages between environmental effects and $^{10}$Be deposition in Lakes Tiefer See and Czechowskie sediments at the end of Chapter 5.1, after describing and performing our procedure used for the attribution and correction of these effects. To be clearer about this sub-structure, we extended and reformulated the introductory paragraph of Chapter 5.1:

'Environment and catchment conditions can add non-production variations to $^{10}$Be$_{con}$ records from varved lake sediments (Berggren et al., 2010; Czymzik et al., 2015). In the following chapter we will, first, describe and perform our approach used for detecting and correcting for possible non-production features in our $^{10}$Be time-series and, then, discuss possible mechanisms behind the statistically inferred connections.'

Page 5 Line 12: The correlations could be added in the figure 4 similarly as is done in figures 2 and 3.

Done.

Page 5 Line 19: Could these depositional mechanisms be briefly described?

See our answer to comment 'Page 4, Line 27'.

Page 5 Line 20-21: At this point it does not become clear which correlations are referred. This
becomes clear later in the paragraph but text would be easier to follow if the correlations were
specified before showing the numbers.

We now mention the correlations between $^{10}$Be and TOC before we provide the
correlation coefficient and significance level (lines 6.5-7).

Page 6 Line 8-9: Why? Are there indications in the sediments that suggest resuspension of
littoral sediments or changes in sediment focusing? The illustration of sediment composition
(see general comments) could be helpful here.

See our Detailed Answer #3 'Visualization of $^{10}$Be and sediment properties'. We added
more detailed information on the sub-layer of resuspended littoral sediments in JC
varves back to 2800 a BP in Section 5.3. To avoid repetition, we prefer not to go into
detail at this point, but to hint more clearly to the later discussion (lines 6.25-29):

'In JC sediments, the $^{10}$Be$_{comp}$ excursion from -50 to 0 a BP (AD 2000-1950) without an
expression in the group sunspot number record as well as the about 20-year delayed
Maunder Minimum response could be explained by transport of 'old' $^{10}$Be from the littoral
to the coring site (see more details on the sub-layer of resuspended littoral sediments in
JC varves back to 2800 a BP in Section 5.3) and/or spatially inhomogeneous $^{10}$Be
deposition patterns (Fig. 5).'

Page 6 Line 29-30: This (also) would be nicely clarified with the record-describing illustration
(see comment for Page 3-4 Methods-Results).

See our Detailed Answer #3 'Visualization of $^{10}$Be and sediment properties'.

Page 7 Line 1-2: What is this layer? Does this occur at the time interval discussed in this paper
at Page 6 Line 6 (from -50 to 0 BP)? If so, this could be mentioned already earlier. This would
actually answer partly to the specific comment I made for Page 6 Line 8-9.

This sub-layer was deposited in fall AD 2003. It consists of the same littoral diatoms and
patches of calcite deposited in Lake Czechowskie during this season since about 2800 a
BP. However, during that year the fall layer was exceptionally thick (3.7 mm), containing
comparably high amounts of 'old' $^{10}$Be leading to anomalous $^{10}$Be concentrations
(Czymzik et al., 2015, *Earth and Planetary Science Letters*). Considering the reviewer's comment, we added more information about this exceptional sediment sub-layer to the
manuscript (lines 7.22-25):

'Comparable influences of sediment resuspension were also found in a sample from an
annually resolved $^{10}$Be time-series of JC sediments covering the period AD 2009-1988
(Czymzik et al., 2015). A varve with an exceptionally thick (3.7 mm) layer of
resuspended littoral diatoms and patches of calcite deposited in fall 2003 reveals
anomalous $^{10}$Be concentrations (Czymzik et al., 2015).'

Figure 4: Why 10Becon and 10Beenvironment are out of phase in Lake Czechowskie
from about 2700 to 3100 BP?

This is an effect of our 'environment correction' procedure. When the correction is large,
the generated signal will look increasingly different from the original $^{10}$Be$_{con}$ record. That
this looks partly like a phase shift is mere coincidence. We discuss in lines 7.17-25 that
we do not obtain significant fits between Lake Czechowskie $^{10}$Be and IntCal13 $^{14}$C
production during the Maunder- and Homeric Minima and point out that this is likely due
to environmental influences on Lake Czechowskie $^{10}$Be, which are challenging to correct
for.
We thank Anonymous Referee #2!